# New classification of superior semicircular canal dehiscence in HRCT

Stephan Waldeck[1,2]*, Heinrich Lanfermann[3], Christian von Falck[4], Matthias F. Froelich[5], René Chapot[6], Marc Brockmann[2], Daniel Overhoff[1,5]

**1** Department of Diagnostic and Interventional Radiology and Neuroradiology, Bundeswehr Central Hospital Koblenz, Koblenz, Germany, **2** Institute of Neuroradiology, University Medical Centre Johannes Gutenberg University Mainz, Mainz, Germany, **3** Institute of Neuroradiology, Hannover Medical School, Hanover, Germany, **4** Institute of Diagnostic and Interventional Radiology, Hannover Medical School, Hanover, Germany, **5** Department of Radiology and Nuclear Medicine, University Medical Centre Mannheim, Medical Faculty Mannheim, Heidelberg University, Mannheim, Germany, **6** Department of Neuroradiology, Alfried Krupp Krankenhaus, Essen, Germany

* dr.waldeck@icloud.com

## Abstract

### Background and purpose

The complex anatomy of the temporal bone is difficult to understand and constitutes a challenge in the daily diagnostic routine even for experienced neuroradiologists. In the context of otoneurological (oVEMP) and preoperative diagnostics, the diagnosis of superior semicircular canal dehiscence (SSCD) is of great importance for Ear, Nose, and Throat (ENT) specialists. The gold standard for this diagnosis is a high-resolution CT (HRCT) of the temporal bone. In order to correctly diagnose SSCD, special oblique reconstructions are necessary in addition to standard (axial, coronal, sagittal) reconstructions. We evaluated the frequency of diagnosis and its location in HRCT in correlation with otoneurological examination. From this analysis, we present a new SSCD classification. This classification yields the potential of a differentiated analysis of the patient's clinical symptoms with correlation to the cross-sectional anatomy and may lead to a differentiated therapy approach.

### Study design and setting

We evaluated 1370 temporal bone scans of patients with residual hearing and verified 343 superior semicircular canal dehiscence (SSCD). We conducted a subgroup analysis of these 343 HRCT scans displaying a SSCD and used them as a basis to create a classification.

### Results

Three location types of SSCD were identified. These were anterior type 1, superior type 2 and posterior type 3. Type 2 were significantly more frequent in both sexes. SSCD at this location can be overlooked if diagnosis is performed only in the standard axial plane, since it can only be visualized by means of double oblique reconstruction. We present a standardized reconstruction algorithm.

**Data Availability Statement:** Data will not be held in a public repository due to the restrictions of the ethics committee. Due to the multicenter data, general publication to protect patient data is not possible and not covered by the ethical vote. Data

can be requested from the corresponding author or from BwZKrhsKoblenzKlinikVIIIRadiologie@bundeswehr.org.

**Funding:** The author(s) received no specific funding for this work.

**Competing interests:** The authors have declared that no competing interests exist.

**Abbreviations:** HRCT, high resolution computed tomography; SSC, superior semicircular canal; SSCD, superior semicircular canal dehiscence; oVEMP, ocular vestibular evoked myogenic potential.

## Conclusion

In total, three types of SSCD with differing incidences can be extrapolated from the locations. Superior type 2 is the most frequent one. Both sexes are affected with roughly equal incidence. The use of standardized double oblique reconstruction algorithm ensures that all three types are diagnosed in the HRCT.

## Introduction

The clinical picture of superior semicircular canal dehiscence (SSCD) was described for the first time by Minor et al. in 1998 [1]. Patients reported vestibular (vertigo, nystagmus, Tullio phenomenon, Hennebert sign [2]) and acoustic signs and symptoms (conductive or sensorineural hearing loss, autophony, tinnitus).

These problems typically occur in nearly all age groups. The average age for the onset of this illness is during middle age (38–61 years) [2–5]. In radiological studies the prevalence has been stated to be between 2.1% and 20.4% [6–9].

Pathophysiologically, the signs and symptoms can be explained by the development of a "third window" caused by an absence of the bone overlying the superior semicircular canal (SSC). This results in a connection between the inner ear and the middle cranial fossa, which is described as a "third window" in addition to the oval and round windows. Pressure changes can trigger endolymph movement in the superior semicircular canal, which can ultimately lead to an excitation of the cupula and cause the described signs and symptoms [10].

The etiology of SSCD is yet to be explained conclusively. Both inflammatory and traumatic causes are being discussed. In the case of trauma, a tearing of the thin superior semicircular canal membrane occurs and the onset of symptoms is instantaneous [11–13]. Fig 1 shows the mechanism that can lead to the symptoms.

High-resolution computed tomography (HRCT) is an essential part of diagnosing SSCD since it offers a high degree of sensitivity [14]. The positive predictive value is given as 93% for a slice thickness of 0.5 mm [15]. Sensitivity decreases dramatically for thicker slices. It is

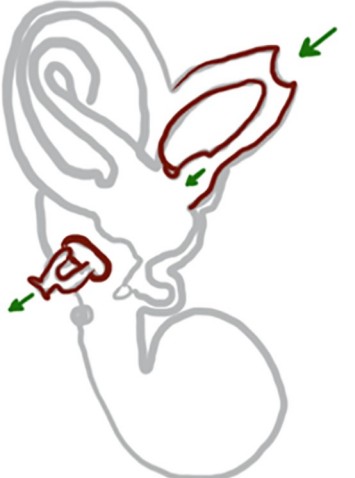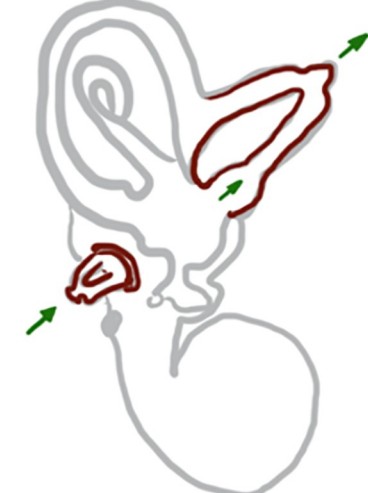

**Fig 1. SSCD symptoms scheme.** The Arrows show the mechanism that can lead to the SSCD symptoms, with both intracranial pressure increases and otoacoustic stimuli acting as possible triggers.

important to understand that SSCD location can differ. With regard to its location in the superior semicircular canal, dehiscence can be anterior, superior or posterior based on the anatomical location. This new classification is introduced by the authors and is based on the radiological cross-sectional anatomy. In order to make sure that all location types are detected, special reconstructions must be generated from the submillimeter slices.

This retrospective multicenter study analyzed 1370 HRCT scans of the temporal bone and verified 343 SSCDs. Their distribution was grouped into three types. The evaluation also included an examination of the incidence of the location types in relation to sex and age.

## Materials and methods

### Study design

A retrospective multicenter study of high-resolution CT scan from patients with vestibular and/or acoustic symptoms. Local ethics committee ("Ethikkommission Landesärztekammer Rheinland-Pfalz") approved this retrospective study (Nr: 2018–13750). For this retrospective study only pseudonymized medical records were used.

Ethics committee waived the requirement for informed consent due to the retrospective nature of this study.

### Patient cohort

We analyzed 1370 HRCT scans of the temporal bone from Ear, Nose and Throat (ENT) patients who presented with vestibular symptoms (from May 2008 to March 2018) for the presence of SSCDs in this multicenter study. For all diagnosed SSCD we performed a subgroup analysis. These were subdivided by gender and grouped into four age groups (A: 6–24, B:25–49, C:50–74, D:75–95 years).

### Data collection

The CT examinations of the temporal bone obtained from the different institutes and departments were conducted based on the following parameters:

Koblenz: Siemens Somatom Force 384-slice scanner; 120 kV; 140 mAs; 0.80 pitch.

Toshiba Vision Edition 320; 120 kV; 200 mAs; 16 cm volume scan

Mainz: Toshiba Aquilon 32-slice scanner; 120 kV; 200 mAs; 0.66 pitch

Hanover: GE LightSpeed 16; 100 kV; 80 mAs; 1.38 pitch

A slice thickness of 0.6 mm was used for multiplanar reconstruction. All data were analyzed with Siemens Syngo Via software (version VB10).

### Standardized reconstruction scheme

In order to detect all SSCDs, the image analysis followed a standardized reconstruction scheme on the basis of the MPRs (multiplanar reconstructions). The image reconstruction and the subgroup analysis of the SSCD localization based on it were carried out in 4 steps:

1. The axial data set is used to generate a reconstruction orthogonal to the superior semi-circular canal with a slice thickness of 0.6 mm (Fig 2A).

2. This reconstruction is used to generate another oblique reformation parallel to the SSC with a slice thickness of 0.6 mm (Fig 2A).

3. The result of the 2 reformations is used to create an MIP (maximum intensity projection) image (1mm) to render the entire SSC visible in one image (Fig 2B).

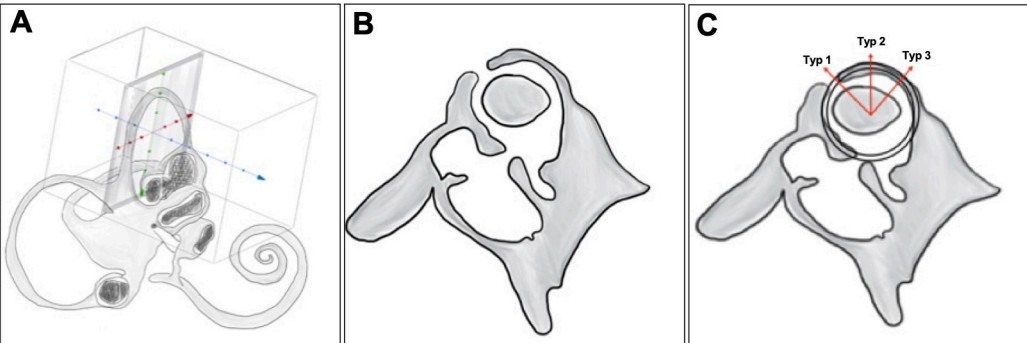

**Fig 2.** A-C: Standardized reconstruction algorithm and SSCD classification: Type 1–3. Fig 2 uses a model to illustrate the reconstruction scheme.

4. The locations of dehiscence in the superior semicircular canal (SSC) were grouped into three types (Fig 2C).

The basis for the classification is the dominant superior type 2 (Fig 2), which is the highest point of circumference the superior semicircular canal (12 o´clock). The borders of this type are defined according to the clock time between 10 am and 2 pm. All superior semicircular dehiscences that lie anteriorly (direction to middle cranial fossa) to this location type 2 are defined as type 1; those that lie posteriorly (direction to posterior cranial fossa) to this location type 2 are definded as type 3.

## Statistical analysis

The statistical calculations were performed using SPSS (IBM SPSS Statistics, version 20.0 for Macintosh; SPSS, Inc., Chicago, IL, USA).

All continuous variables are expressed as arithmetic mean ± standard deviation (SD). A significance level of 5% was used. Normal distribution was assumed due to central limit theorem. A multinominal linear regression was performed type-specific SSCD proportions as dependent variable and gender, side of SSCD and age as covariates. For continuous data Student's t-Test was performed for independent samples to analyze the means. To analyze nominal data Chi- square test was used.

## Results

We analyzed 1370 high resolution CT scans of the temporal bone from ENT patients (681 female (49.7%) versus 689 male (50.3%) for the presence of SSCDs and verified 343 diagnoses out of 280 patients (prevalence of 20.4%). Patients with SSCD were on average 6 years older than those without SSCD (mean 46.22 years versus 52.73 years (SSCD) p<0.001).

63 patients had a diagnosis of bilateral SSCD, 116 unilateral left, 101 unilateral right; unilateral/bilateral ratio of 3.44:1. Table 1 summarizes all the results of the gender in relation to the frequency of SSCD types and its locations.

We were able to visualize all SSCD types with the standardized reconstruction scheme (Fig 3).

The total of 280 patients with SSCD (123 female and 157 male; p = 0.030) were grouped into four age groups (A:6–24, B:25–49, C:50–74, D:75–95 years).

Of a total of 343 SSCD, 99 (28.9%) displayed dehiscence of an anterior location type (type 1), 133 (38.8%) displayed the dominant superior location type (type 2) and 111 (32.4%) displayed the posterior location type (type 3). The different Types are shown in Fig 4.

**Table 1. Study population.**

| Population n = 1370 | SSCD | | |
|---|---|---|---|
| **sex** | **male** | *female* | |
| | 157 (11.5%) | *123 (9.0%)* | |
| **side** | *right* | *left* | **both** |
| | 101 (7.4%) | 116 (8.5%) | 63 (4.6%) |
| **type n = 2740** | **1** | **2** | **3** |
| | 99 (3.6%) | 133 (4.9%) | 111 (4.1%) |
| | **male/ female** | **male/ female** | **male/ female** |
| | 56(2.0%)/43(1.6%) | 78(2.8%)/55(2.0%) | 57(2.1%)/54(2.0%) |

In brackets the relative values compared to the total study population; Superior semicircular canal type with n = 2740 resulting from both temporal bones of each patient (n = 1370).

SSCD type 2 is significantly more common in males (p = 0.048) and significantly more common in age group C (p = 0.012) and D (p = 0.004), whereas the multinominal linear regression showed just a tendency with p = 0.064 for sex without significancy.

The multinominal linear regression was significant for age and not for side of the SSCD location.

SSCD type 1 and SSCD type 3 are not significantly more common in all age groups.

Overall, no significant difference was evident between the sexes in terms of incidence, as shown by the multinominal linear regression model.

The age-correlated increase of SSCDs lead to a significant change in the location of dehiscence. There was a dominance of the superior location of dehiscence corresponding to type 2 in 3 of the 4 age groups (group 2: 25–49 y; group 3: 50–74 y; group 4: 75–95 y). Across all age groups, SSCD type 2 is significantly more common than type 3 and than type 1 (type 2 > type3 > type1).

Table 2 summarizes all the results of the different age groups in relation to the frequency of SSCD types.

## Discussion

Superior Semicircular Canal Dehiscence (SSCD) is still a relatively new pathologic condition that requires a high suspicion for its correct diagnosis and high quality CT scan imaging. While different surgical approaches and techniques have been published in the literature, the exact location of the SSCD is of high importance. To the best of our knowledge the presented analysis includes a considerably larger number of cases than other published radiological studies before [8]. We are the first to analyze the location of superior semicircular canal dehiscence

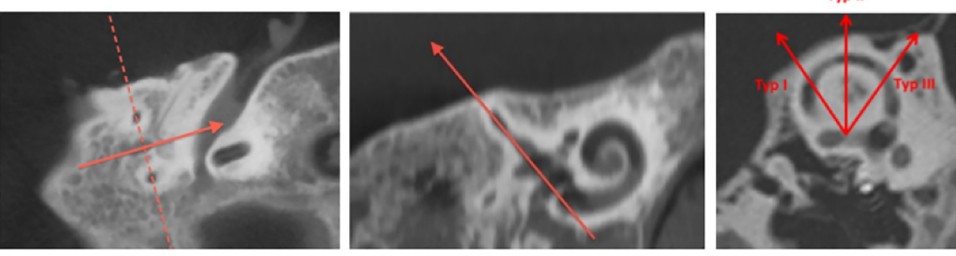

**Fig 3. HR CT reconstruction.** Fig 3 illustrates the practical implementation with a temporal bone preparation.

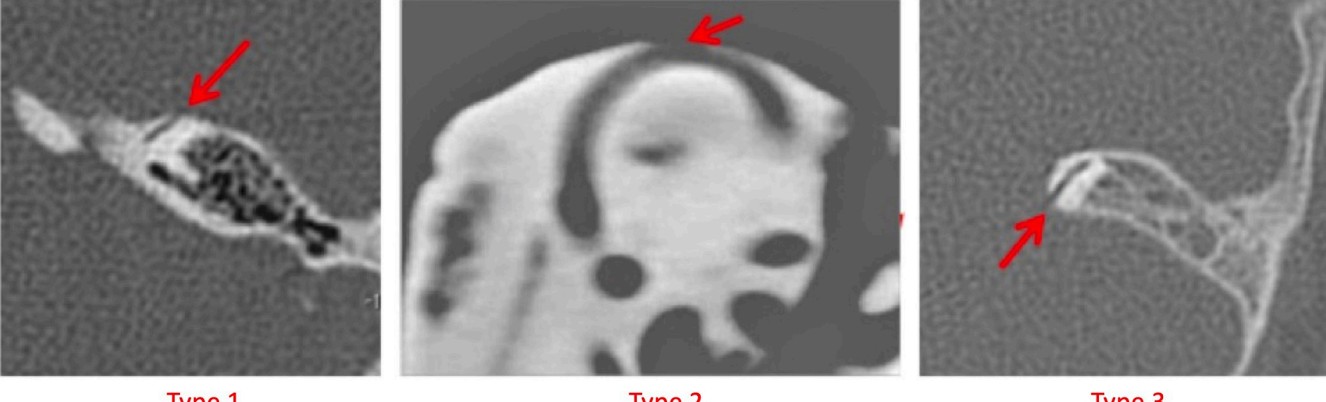

**Fig 4. SSCD location types.** Fig 4 shows the three different SSCD Types in high resolution CT. Type 2 SSCDs are defined as the highest point of the circumference the superior semicircular canal. All SSCDs that lie anteriorly in direction to the middle cranial fossa are defined as type 1; those that lie posteriorly in direction to the posterior cranial fossa are definded as type 3.

based on a fixed reconstruction scheme. CT is still said to result in an overdiagnosis of SSCD, especially for slice thicknesses of $\geq$ 1 mm [7]. In our experience, significant misdiagnosis occurs when attempting to diagnose SSCD with CT slice thicknesses greater than 1 mm. Furthermore, we consider it imperative to verify the SSCD diagnosis multiplanar, otherwise overdiagnosis or misdiagnosis may occur. Nevertheless the high resolution CT, it is an integral part of equipment-based diagnostics of SSCD in case of clinical suspicion. In the future, newer ultra high resolution cross-sectional imaging techniques, such as using photon counting CT (UHR PCD-CT), have the potential to provide even better visualizations of SSCD at low radiation dose [16].

In our study all multiplanar reconstructions were evaluated based on sub-millimeter slices and complemented by standardized double-oblique reconstruction [17]. Our patient group displayed a relative dominance of the type 2 location regardless of sex and age. This SSCD type 2 can be overlooked in standard axial reconstructions because it is in the superior semicircular canal apex. We were able to reliably visualize SSCD location type 2 in all cases using the reconstruction algorithm presented here.

343 SSCD data sets from 280 patients were analyzed for our subgroup analysis. The data originated from an analysis of 1370 petrous temporal bone HRCTs. At a prevalence of up to 20.42%, SSCD is a new but not rare diagnosis in patients with residual hearing. The results largely correspond to those of other radiological studies (Nadgir et al.: 7.9% (n = 304) [18]; Stimmer et al.: 12.6% (n = 350) [8]; Williamson et al.: 13.5% (n = 223) [7]; Ceylan et al.: 17.5% (n = 108) [9]).

**Table 2. Differentiation of SSCD types by age cohorts.**

|                | 0–24 yo    | 25–49 yo    | 50–74 yo     | >75 yo      | all ages     |
|----------------|------------|-------------|--------------|-------------|--------------|
| SSCD Type 1    | 19 (2.7%)  | 26 (3.5%)   | 27(3.3%)     | 27(5.0%)    | 99(7.2%)     |
| SSCD Type 2    | 26 (3.7%)  | 32 (4.3%)   | 39 (5.2%)    | 36(6.7%)    | 133(9.7%)    |
| SSCD Type 3    | 20(2.8%)   | 25(3.4%)    | 36 (5.0%)    | 30(5.6%)    | 111(8.1%)    |
| SSCD Type1-3   | 65(9.2%)   | 83(11.2%)   | 102(13.5%)   | 93(17.2%)   | 343(25.0%)   |

Relative values in percent in parentheses. The relative values of the age cohort columns refer to the respective age cohort. The relative values of the "all ages" columns refer to the total study population.

It is important to know about and correctly diagnose SSCD. Our reconstruction scheme notably reduces the risk of overdiagnosis when using CT. The risk of overdiagnosis is especially large if slice thicknesses of $\geq 1$ mm are used. Belden et al. demonstrated the importance of selecting a slice thickness of $< 1$ mm in diagnosing SSCD when they disproved in all 36 cases an initial suspicion of SSCD formed after acquisition based on 1 mm slice thickness by re-examining the cases with 0.5 mm slices [15]. We therefore recommend the use of a scan protocol with a slice thickness of $\leq 0.6$ mm.

The present retrospective analysis showed overall age groups no significant difference in the relative incidence of SSCD for women and men. In the subgroup analysis SSCD type 2 is significantly more common in males and significantly more common in age group C and D. These results are in line with the observations made by Crovetto et al. [19], da Cunha Ferreira et al. [20] and Martin et al. [21].

## Study limitations

Our retrospective study focused on the incidental findings of SSCD in the HRCT imaging as primary outcome. A multiparametric analysis with respect to clinical parameters such as vestibular symptoms or in the correlation of new high frequency ocular vestibular evoked myogenic potential (oVEMP n10 response) for the study cohort were not the primary target of this study. Clinical parameters should be analyzed in another prospective study approach to demonstrate the clinical added value of the classification of SSCD. Statistically reliable investigations about the different SSCD types in comparison with the clinical parameters remain to be conducted in a prospective study.

## Conclusions

From the neuroradiological point of view, the relatively new SSCD diagnosis with its prevalence of up to 20.42% among individuals with residual hearing is not a rare one. This makes it even more important for the radiologist to obtain certainty by confirming the diagnosis by means of HRCT with slice thickness under 0.6 mm. Our reconstruction scheme makes a reliable visualization and clear classification of SSCD possible. This standardization yields the potential of a correlation of the patient's clinical symptoms with cross-sectional anatomy and may lead to a differentiated therapy approach.

## Author Contributions

**Conceptualization:** Stephan Waldeck, Heinrich Lanfermann, Christian von Falck, Matthias F. Froelich, René Chapot, Marc Brockmann, Daniel Overhoff.

**Data curation:** Stephan Waldeck, Heinrich Lanfermann, Christian von Falck, Matthias F. Froelich, René Chapot, Marc Brockmann, Daniel Overhoff.

**Formal analysis:** Stephan Waldeck, Heinrich Lanfermann, Christian von Falck, Matthias F. Froelich, René Chapot, Marc Brockmann, Daniel Overhoff.

**Funding acquisition:** René Chapot.

**Investigation:** Stephan Waldeck, Heinrich Lanfermann, Matthias F. Froelich, René Chapot, Daniel Overhoff.

**Methodology:** Stephan Waldeck, Heinrich Lanfermann, Christian von Falck, Matthias F. Froelich, Marc Brockmann, Daniel Overhoff.

**Project administration:** Stephan Waldeck, Heinrich Lanfermann, Daniel Overhoff.

**Resources:** Stephan Waldeck, Matthias F. Froelich, Daniel Overhoff.

**Software:** Stephan Waldeck, René Chapot, Marc Brockmann, Daniel Overhoff.

**Supervision:** Stephan Waldeck, Heinrich Lanfermann, Christian von Falck, Matthias F. Froelich, René Chapot, Marc Brockmann, Daniel Overhoff.

**Validation:** Stephan Waldeck, Heinrich Lanfermann, Christian von Falck, Matthias F. Froelich, Marc Brockmann, Daniel Overhoff.

**Visualization:** Stephan Waldeck, Christian von Falck, Matthias F. Froelich, René Chapot, Marc Brockmann, Daniel Overhoff.

**Writing – original draft:** Stephan Waldeck, Heinrich Lanfermann, Christian von Falck, Matthias F. Froelich, René Chapot, Marc Brockmann, Daniel Overhoff.

**Writing – review & editing:** Stephan Waldeck, Heinrich Lanfermann, Christian von Falck, Matthias F. Froelich, René Chapot, Marc Brockmann, Daniel Overhoff.

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
