## [Decision Letter · Decision Letter 0]

10 Dec 2021

PONE-D-21-25618

New classification of superior semicircular canal dehisence in HRCT

PLOS ONE

Dear Dr. Waldeck,

Thank you for submitting your manuscript to PLOS ONE. After careful consideration, we feel that it has merit but does not fully meet PLOS ONE’s publication criteria as it currently stands. Therefore, we invite you to submit a revised version of the manuscript that addresses the points raised during the review process.

We look forward to receiving your revised manuscript.

Kind regards,

Jorge Spratley, MD, PhD

Academic Editor

PLOS ONE

Additional Editor Comments (if provided):

Dear Authors,

Thank you for having submtted your original manuscript to PlosOne and for patiening for the decision.

Please carefully address and reply to all reviewers comments and questions, as appropriate.

Also please take into consideration the following remarks:

1. Your citations on the text do not match the References list (see last paragraph of discussion). Please correct.

2. The limitations of CT scan in diagnosing SSCC dehiscence should be discussed in a more comprehensive fashion, in particular on the risk of imaging overdiagnosis.

Journal Requirements:

Reviewers' comments:

Reviewer's Responses to Questions

**Comments to the Author**

1. Is the manuscript technically sound, and do the data support the conclusions?

Reviewer #1: Yes

Reviewer #2: No

Reviewer #3: Yes

2. Has the statistical analysis been performed appropriately and rigorously? 

Reviewer #1: I Don't Know

Reviewer #2: No

Reviewer #3: N/A

3. Have the authors made all data underlying the findings in their manuscript fully available?

Reviewer #1: Yes

Reviewer #2: No

Reviewer #3: Yes

4. Is the manuscript presented in an intelligible fashion and written in standard English?

Reviewer #1: Yes

Reviewer #2: Yes

Reviewer #3: Yes

5. Review Comments to the Author

Reviewer #1: General Comments

The study submitted to PLOSONE presents the results of an original research study of superior semicircular canal dehiscence (SCCD), after an extensive review of 1370 temporal bones. Altogether, It tries to overcome one of the major problems on SSCD diagnosis which is its common over diagnostic rate.

The tests, statistics, and other analyses performed are described in detail.

Conclusions are presented in an appropriate fashion and are supported by the data.

The article is presented in an intelligible fashion and is written in standard English, however with some minor errors.

The research meets all applicable standards for the ethics of experimentation and research integrity.

The article adheres to appropriate reporting guidelines and community standards for data availability.

The paper globally shows how a standardized reconstruction scheme on CT can improve the diagnosis and help a more accurate clinical approach on the involved pathology.

Specific comments

It would have been interesting and very much clinically valuable to have compared the CT results with the vestibular myogenic potentials results. A simple exam that apports decisive information of SSCD. If available, that information and analysis should be added to the evaluation.

Minor errors in the text:

und (page 4)

Initials MPR and MIP are not presented previously (page 5)

Legend Table 1. sex relative values to the sex; side, type relative values to total population

Reviewer #2: PONE-D-21-25618: statistical review

SUMMARY. This study relies on a large sample (n=1370) of high-resolution scans of the temporal bone in ENT patients. The principal outcome is the proportion of cases of superior semicircular canal dehiscence (SSCD), clustered according to three location types. The statistical analysis compares these proportions between genders and across age groups. Although the relationship between SSCD proportions and relevant clinical parameters of the patients would have been much more interesting, the authors explicitly declare that this was not the target of the study. With this limitation, the study reduces to an epidemiological study of SSCD incidence across demographic groups. I list below two major issues that should be addressed and three specific points that should be clarified.

MAJOR ISSUES

1) Little information is provided about the sample. Do the subjects share specific characteristics? Is it a multi-center study? What were the reasons for visiting these patients? Without these information, we are unable to evaluate whether the results of the study can be extended to a target population.

2) The statistical tools are below the level required by the Journal. As the purpose of the study is the comparison of proportions across age and gender groups, the natural approach is a multinomial regression where type-specific SSCD proportions is the outcome and age and gender are the two regressors. Multinomial regression is also able to test a possible interaction effect between age and gender. The model is available in SPSS, which is the software used by the authors.

SPECIFIC ISSUES

1) Page 5. “Normal distribution was assumed due to central limit theorem”. This sentence is a bit obscure. Do you want to explain why proportions are being compared by a t-test?

2) Page 5. “For continuous data Student`s t- Test was performed for independent samples to analyze the means”. I don’t see continuous data in your study: you have proportions, age groups (taken as factors) and gender. Please clarify.

3) Page 5. “To analyze ordinal data Chi- square test was used.” Similar to above: where are the ordinal data in this paper? In addition: I'm not aware of a Chi Square test for ordinal data: could you please specify the test that has bneen used?

Reviewer #3: the subject in the article is an interesting approach to SSCD and may contribute to better diagnosis and understanting of the disease. about the classification in type 1, 2 or 3 a better explanation about the imaging criteria is needed.

the authors need to review also one of the tables.

The article may be accepted with minor review

6. PLOS authors have the option to publish the peer review history of their article (what does this mean?). If published, this will include your full peer review and any attached files.

Reviewer #1: No

Reviewer #2: No

Reviewer #3: No

---

## [Author Response · Author response to Decision Letter 0]

27 Dec 2021

Additional Editor Comments (if provided):

Dear Authors,

Thank you for having submitted your original manuscript to PlosOne and for patiening for the decision.

Please carefully address and reply to all reviewers comments and questions, as appropriate.

Also please take into consideration the following remarks:

1. Your citations on the text do not match the References list (see last paragraph of discussion). Please correct.

We thank the editor for this comment and corrected the passage accordingly. In this case there was no citation in-tended at this passage.

2. The limitations of CT scan in diagnosing SSCC dehiscence should be discussed in a more comprehensive fash-ion, in particular on the risk of imaging overdiagnosis.

Thank you very much for this suggestion, which we have gladly implemented. In the discussion, we more clearly highlighted the risk of overdiagnosis by monoplanar CT with 1 mm thick slices and made a recommendation for high-resolution imaging.

Journal Requirements:

https://journals.plos.org/plosone/s/file?id=wjVg/PLOSOne_formatting_sample_main_body.pdfand

We made the changes according to the journal style requirements.

2. In your Data Availability statement, you have not specified where the minimal data set underlying the results de-scribed in your manuscript can be found. PLOS defines a study's minimal data set as the underlying data used to reach the conclusions drawn in the manuscript and any additional data required to replicate the reported study findings in their entirety. All PLOS journals require that the minimal data set be made fully available. For more information about our data policy, please see http://journals.plos.org/plosone/s/data-availability.

Upon re-submitting your revised manuscript, please upload your study’s minimal underlying data set as either Support-ing Information files or to a stable, public repository and include the relevant URLs, DOIs, or accession numbers within your revised cover letter. For a list of acceptable repositories, please see http://journals.plos.org/plosone/s/data-availability#loc-recommended-repositories. Any potentially identifying patient information must be fully anonymized.

Important: If there are ethical or legal restrictions to sharing your data publicly, please explain these restrictions in detail. Please see our guidelines for more information on what we consider unacceptable restrictions to publicly shar-ing data: http://journals.plos.org/plosone/s/data-availability#loc-unacceptable-data-access-restrictions. Note that it is not acceptable for the authors to be the sole named individuals responsible for ensuring data access.

We agree with the Journal that data sharing is an integral part of good scientific work. Unfortunately, for ethical reasons and the ethics committee's vote, even anonymized patient data cannot be shared. But data sets are available upon request.

Reviewers' comments:

Reviewer #1: General Comments

The study submitted to PLOSONE presents the results of an original research study of superior semicircular canal dehiscence (SCCD), after an extensive review of 1370 temporal bones. Altogether, It tries to overcome one of the major problems on SSCD diagnosis which is its common over diagnostic rate.

The tests, statistics, and other analyses performed are described in detail.

Conclusions are presented in an appropriate fashion and are supported by the data.

The article is presented in an intelligible fashion and is written in standard English, however with some minor errors.

The research meets all applicable standards for the ethics of experimentation and research integrity.

The article adheres to appropriate reporting guidelines and community standards for data availability.

The paper globally shows how a standardized reconstruction scheme on CT can improve the diagnosis and help a more accurate clinical approach on the involved pathology.

Specific comments

It would have been interesting and very much clinically valuable to have compared the CT results with the vestibular myogenic potentials results. A simple exam that apports decisive information of SSCD. If available, that information and analysis should be added to the evaluation.

Thank you very much for this recommendation. Due to the retrospective multicenter analysis, these clinical data were not available and could not be provided. We will be happy to include this recommendation in a prospective approach. We have added this information to the text under study limitations.

Minor errors in the text:

und (page 4)

We thank the reviewer for this hint. We corrected that mistake.

Initials MPR and MIP are not presented previously (page 5)

We added this information to the manuscript.

Legend Table 1. sex relative values to the sex; side, type relative values to total population

We have restructured the table legend and hope that the table is now easier to interpret.

Reviewer #2: PONE-D-21-25618: statistical review

SUMMARY. This study relies on a large sample (n=1370) of high-resolution scans of the temporal bone in ENT pa-tients. The principal outcome is the proportion of cases of superior semicircular canal dehiscence (SSCD), clustered according to three location types. The statistical analysis compares these proportions between genders and across age groups. Although the relationship between SSCD proportions and relevant clinical parameters of the patients would have been much more interesting, the authors explicitly declare that this was not the target of the study. With this limitation, the study reduces to an epidemiological study of SSCD incidence across demographic groups. I list below two major issues that should be addressed and three specific points that should be clarified.

MAJOR ISSUES

1) Little information is provided about the sample. Do the subjects share specific characteristics? Is it a multi-center study? What were the reasons for visiting these patients? Without these information, we are unable to evaluate whether the results of the study can be extended to a target population.

Thank you very much for these excellent questions. We have incorporated all the requested information in various places in the text. Here is a brief summary again. This is a multicenter study. All patients presented you with different vestibular symptoms (like vertigo, nystagmus, Tullio phenomenon) in the three participating clinics and received high resolution CT imaging of the petrous bone for clarification in the context of these symptoms.

2) The statistical tools are below the level required by the Journal. As the purpose of the study is the comparison of proportions across age and gender groups, the natural approach is a multinomial regression where type-specific SSCD proportions is the outcome and age and gender are the two regressors. Multinomial regression is also able to test a possible interaction effect between age and gender. The model is available in SPSS, which is the software used by the authors.

We are grateful for the reviewer's comment. We have accordingly calculated a

multinomial linear regression with type-specific SSCD proportions as dependent variable and gender, side of SSCD and age as covariates. We added the fact to the manuscript accordingly.

SPECIFIC ISSUES

1) Page 5. “Normal distribution was assumed due to central limit theorem”. This sentence is a bit obscure. Do you want to explain why proportions are being compared by a t-test?

The central limit theorem states that above a certain size, specified in the literature as 30, the distribution of a popula-tion can be assumed to be bell-shaped and the normal distribution can therefore be taken as given. It would certainly have been possible to test for a normal distribution for the t-test, but we considered this unnecessary for the very large study population.

2) Page 5. “For continuous data Student`s t- Test was performed for independent samples to analyze the means”. I don’t see continuous data in your study: you have proportions, age groups (taken as factors) and gender. Please clari-fy.

We thank the reviewer for this remark. Age was examined as an independent category from the subsequent classifica-tion into age groups. This was also stated in the results:

“Patients with SSCD were on average 6 years older than those without SSCD (mean 46.22 years versus 52.73 years (SSCD) p<0.001).”

Although there is some discourse in the literature as to whether age should be considered continuous or discrete, we believe it is acceptable to refer to it as continuous data. We think that this will also be approved by the reviewer.

3) Page 5. “To analyze ordinal data Chi- square test was used.” Similar to above: where are the ordinal data in this paper? In addition: I'm not aware of a Chi Square test for ordinal data: could you please specify the test that has bneen used?

We thank the reviewer for the excellent comment. This is an error on our part. What was meant here, of course, was nominal data (age, gender, side). This also explains the answer to the question about the choice of the statistical test "Chi-square test", which is permissible for nominal data. We have corrected this fact

Reviewer #3: the subject in the article is an interesting approach to SSCD and may contribute to better diagnosis and understanding of the disease. about the classification in type 1, 2 or 3 a better explanation about the imaging criteria is needed.

Thank you very much for this recommendation which we have implemented very gladly in the text. We have added the comprehensive explanation under Figure 2.

The authors need to review also one of the tables.

We corrected table 1 and the legend of table 1.

---

## [Editor Report · Decision Letter 1]

5 Jan 2022

New Classification of superior semicircular canal dehiscence in HRCT

PONE-D-21-25618R1

Dear Dr. Waldeck,

We’re pleased to inform you that your manuscript has been judged scientifically suitable for publication and will be formally accepted for publication once it meets all outstanding technical requirements.

Kind regards,

Jorge Spratley, MD, PhD

Academic Editor

PLOS ONE

Additional Editor Comments (optional):

Thank you for the sound revision and improvement of the manuscript.

Again, I would like to express our appreciation for having patiented for the referees' revisions.

The paper has now reached the level to be published at PlosOne. Congratulations!
---

## [Editor Report · Acceptance letter]

7 Jan 2022

PONE-D-21-25618R1 

New Classification of superior semicircular canal dehiscence in HRCT 

Dear Dr. Waldeck:

I'm pleased to inform you that your manuscript has been deemed suitable for publication in PLOS ONE. Congratulations! Your manuscript is now with our production department. 

Kind regards, 

on behalf of

Professor Jorge Spratley 

Academic Editor

PLOS ONE